# The Effect of Dividend Payment on Firm's Financial Performance: An Empirical Study of Vietnam

**Anh Huu Nguyen, Cuong Duc Pham \*, Nga Thanh Doan, Trang Thu Ta, Hieu Thanh Nguyen and Tu Van Truong**

School of Accounting and Auditing, The National Economics University, Hanoi 113068, Vietnam;
anhh@neu.edu.vn (A.H.N.); doanthanhng@neu.edu.vn (N.T.D.); trangt@neu.edu.vn (T.T.T.);
hieuketoa@neu.edu.vn (H.T.N.); Truongt@neu.edu.vn (T.V.T.)
\* Correspondence: cuongpd@neu.edu.vn

**Abstract:** This research aims to investigate the effects of dividend policies on a firms' financial performance. The paper explores the research gap and then builds a research model using ROA, ROE, and Tobin's Q as dependent variables, dividend rate and decision of dividend payment as independent variables. The paper collected data and financial statements of 450 firms that are listing on the stock market of Vietnam from 2008 to 2019. The analysis results indicate that the decision of dividend payment has negative impact to Vietnamese firms measured by accounting-based performance but this improve market expectation on firms. In addition, the paper finds that Vietnamese firms are offering low dividend rate which has a positive impact on accounting-based performance but a negative effect on market expectation. This paper proposes some instructive recommendations based on the findings, including a more appropriate model of dividend policies, a lower dividend rate, and clear decision of dividend payment.

**Keywords:** dividend rate; decision of dividend payment; financial performance; Vietnam

**JEL Classification:** M41; G32; D82

## 1. Introduction

A dividend is a distribution of profits by a corporation to its shareholders. When a corporation earns a profit or surplus, it is able to pay a proportion of the profit as a dividend to shareholders. The remaining profit after dividend, namely retained earnings will be used to re-invest in the future.

A high dividend payment means that the company is reinvesting less money back into its business. According to Khan et al. (2019) companies with high dividend tend to attract investors who prefer the assurance of a steady stream of income to a high potential for growth in the share price. On the contrary, companies with low dividend payment means that the companies is reinvesting in business growth, so that the higher future capital gains for investors.

Studies on the impact of the dividend policy on the financial performance of enterprises have been carried out by many scholars around the world. Some scholars believed that this topic is one of the most challenging research issues (Onanjiri and Korankye 2014; Frankfurter and Wood 2002; Amidu 2007). Some others think that dividend policy is not only business transaction, but it is firm's strategy applied to distribute income to shareholders (for instance, Gill et al. 2010).

Regardless of various research, the previous studies have evidenced the differences about the impact of dividend policy on a firm's financial performance. Some scholars believed that dividend policy significantly and positively impact on financial performance (for instance, Ali et al. 2015). Some others reported that dividend policy impact significantly but negatively to firm performance (Onanjiri and Korankye 2014). The differences in research result are not only between research years but also inconsistent across countries

(Glen et al. 1995; Kim and Kim 2020), and even among economic sectors in a specific country (Khan et al. 2019).

This paper is motivated by occurrence of different research results. It aims to find the effects of dividend policy (represented by dividend rate and decision of dividend payment) on firm's financial performance. Findings from this could contribute to literature by confirming the previous research. Moreover, it could help firms' managers in setting dividend policies for listed firms through which could control their cashflow and financial performance.

The paper has the following parts: the next section is the analysis of the literature review and theoretical framework, followed by the research methodology, empirical findings, discussions and recommendations, and concluding remarks.

## 2. Literature Review and Theoretical Framework

Firms' management always concerns about whether to pay dividends to shareholders or to retain them for future re-investments and what percentage should be applied if dividend payments are made. The firms' managers need to align shareholders, insiders, and outsiders with each other.

The theoretical principle underlying the effect of dividend policy on a firm's performance can be described in dividend relevance theory stated by Miller and Modigliani (1961). They theorized that dividend policy has no impact on stock price and cost of capital, resultantly the dividend policy of a firm is irrelevant for shareholders' wealth in keeping with perfect capital market assumptions. Miller and Modigliani (1961) revealed a well-designed analysis of the relationship between dividend policy, growth, and share valuation. Based on well-defined but a simplified set of perfect capital market assumptions, Miller and Modigliani (1961) expressed a dividend irrelevance theorem. According to this concept, investors do not pay any importance to the dividend history of a company and thus, dividends are irrelevant in calculating the valuation of a company. Due to the distribution of dividends, the price of the stock decreases and will nullify the gain made by the investors because of the dividends.

Litzenberger and Ramaswamy (1982) showed that dividend policy influences investor behaviors as a result of disparity in taxation on dividends and capital gains. The authors believed that investors prefer low-dividend businesses since the amount of taxes payable is minimized. Jensen and Meckling (1976) stated that there is a tradeoff in the form of agency costs between having more or less insider ownership. Agency costs are created whenever the manager also controls an outsider's investment besides her own, because there is a fundamental conflict of interest.

The next underlying theory is the pecking order theory which was first introduced by Donaldson (1961) and modified by Myers and Majluf (1984). The Pecking Order Theory relates to a company's capital structure. The theory states that managers follow a hierarchy when considering sources of financing. The pecking order theory arises from the concept of asymmetric information which causes an imbalance in transaction power. Company managers typically possess more information regarding the company's performance, prospects, risks, and future outlook than external users such as creditors (debt holders) and investors (shareholders). Therefore, to compensate for information asymmetry, external information users demand a higher return to counter the risk that they are taking. As opposed to external financing, internal financing is the cheapest and most convenient source of financing.

Another underlying theory for dividend policies is the signaling theory that was firstly introduced by Spence (1973) and it is useful for describing behavior when two parties (individuals or organizations) have access to different information sources as sender or receiver and both parties act differently. Dividend signaling is a theory that suggests that company announcements of dividend increases are an indication of positive future results. Increases in a company's dividend payout generally forecast a positive future performance of the company's stock. In the finance area, the reporting principle shows that the shift in

dividends will give shareholders an indication of the future profitability of the business and perceptions of management. Management will not increase dividends unless it is certain that future earnings will meet the dividend increase. The decline in dividend payout is considered a negative signal because investors will think that the company's future earnings are going to decrease (Miller 1980).

Amidu (2007) examined whether dividend policy influences firm performance in Ghana. The analyses are performed using data derived from the financial statements of listed firms on the Ghana Stock Exchange (GSE) during the recent eight-year period. The results showed positive relationships between return on assets, dividend policy, and growth in sales. Surprisingly, the study reveals that bigger firms on the GSE perform less with respect to return on assets. The results also revealed negative associations between return on assets and dividend payout ratio. The results of the study generally supported previous empirical studies (see, for instance, Michaely and Allen 2002; Gordon 1963; Bhattacharya 1979; Shefrin and Statman 1984; Easterbrook 1984; Amidu and Abor 2006; Danila et al. 2020).

Murekefu and Ouma (2012) sought to establish the relationship between dividend policy with dividend payout as representative variable and firm performance among listed firms in the Nairobi Securities Exchange. The findings indicated that dividend payout was a major factor significantly positive affecting firm performance. It can be concluded, that dividend policy is relevant and that managers should devote adequate time in designing a dividend policy that will enhance firm performance and therefore shareholder value.

Also using dividend payout variable, Onanjiri and Korankye (2014) ascertained the impact of dividend policy on the financial performance of manufacturing firms that are trading on the Ghana Stock Exchange. The regression results reveal that dividend payout significantly negative impacts firms' financial performance. For the control variables, size and leverage were inversely related to performance while sales growth positively correlated with performance. Except for size, all the control variables were found to be statistically significant. Intuitively, quoted manufacturing firms in Ghana which are interested in accentuating their return on assets may have to rationalize the quantum of dividend payout. This will help them accumulate high retained earnings to buttress investment in positive net present value projects which will fuel sales growth and thereby lessen their dependence on expensive debt finance in Ghana.

With the same vein of research, Velnampy et al. (2014) attempted to find out the relationship between dividend policy and firm performance of listed manufacturing companies in Sri Lanka. A set of listed manufacturing companies was investigated. Returns on equity and return on assets were used as the determinants of firm performance whereas dividend payout and earnings per share were used as the measures of dividend policy. The study found that determinants of dividend policy are not correlated to the firm performance measures of the organization.

Using dividend payment as independent variable for the model testing the influence of dividend on financial performance of 172 firms in Istanbul Stock Exchange, Dogan and Topal (2014) showed that dividend payments had influence on companies' performances, but in different ways for accounting-based indicators (ROA, ROE) and for market-based one (Tobin's Q).

Vijayakumaran and Atchyuthan (2017) empirically examined the relationship between cash holdings and corporate performance using a sample of firms listed on the Colombo Stock Exchange over the period 2011–2015. Controlling for unobserved heterogeneity and other firm characteristics, this study found that cash holdings are positively related to firm performance. Ali et al. (2015) attempted to find out the impact of dividend policy on firm performance under high or low debt for all the non-financial sector companies listed on the Karachi Stock Exchange. They utilized the secondary data published by the State Bank of Pakistan in the shape of balance sheet analysis of the non-financial sector from 2006 to 2011 with the sample size consisting of 122 companies. They mainly focused on using two performance measures i.e., Tobin's Q and Return on Equity both as dependent

variables while the control variable includes the firm size and growth with debt as the moderating variable. They found that the dividend payout ratio has a significant positive relationship with Tobin's Q and ROA when there is both less and high debt. Besides, there is no moderating effect of debt on the relationship between dividend payout ratio and firm performance of all the non-financial firms listed on the KSE.

Khan et al. (2016) conducted a similar study for some firms listed on the Pakistan Stock Exchange from 2010 to 2015. The OLS technique was applied and the research results evidence that there is a positive relation between return on assets, dividend policy, and sale growth. The results of the research are mostly similar to those of Ali et al. (2015). Specifically, the results show that the dividend payout ratio and leverage have a significant negative relation with the return on equity.

M'rabet and Boujjat (2016) sought to examine the relationship between dividend policies and financial performance of selected listed firms in Morocco. Two models were developed in an attempt to provide a theoretical explanation on the birds-in-hand dividend relevance theory and the Miller and Modigliani's (1961) dividend irrelevance theory. The findings indicated that dividend policy is an important factor affecting firm performance. Their relationship was also significantly positive. This, therefore, showed that dividend policy was relevant. It can be concluded, based on the findings of this research that dividend policy is relevant and that managers should devote adequate time in designing a dividend policy that will enhance firm performance and therefore shareholder value.

In Vietnam, the dividend payment of many companies listed on the stock market, without any strategic credits, is still spontaneous. Regarding dividend payment, businesses would have different ways of dividend payout at different times. Businesses pay dividends more often with higher payout ratios in certain industries when the business operation is profitable. Proponents of dividends point out that a high dividend payout is important for investors because dividends provide certainty about the company's financial well-being. However, the dividend may not be paid, even though businesses are profitable. If a company thinks that its own growth opportunities are better by available investment opportunities elsewhere, it often keeps the profits and reinvests them into the business. When a company decides not to offer a dividend payment, it keeps more money for its own operations. Instead of rewarding investors with a payment, it can invest in its operations or fund expansion in hopes of rewarding investors with more valuable shares of a stronger company. Research on this issue has been conducted, but the results are generally applicable to manufacturing companies (Tran et al. 2015).

In Vietnam circumstance, there have been few studies about the relationship between dividend policy and a firm's performance that report similar results. Tran et al. (2015) applied the previously accepted model with ROE, ROA, and Tobin's Q are dependent variables, and dividend payout ratio and decision of dividend payment (binary value) are independent variables. The data was collected from audited financial statements of listed firms from 2009–2013. The author report that the cash dividend payment has a significant impact on the firm's performance measured. However, the dividend payout ratio has negative impact on firms' financial performance. This research contributes significantly to literature, however, this study contained some drawbacks, such as (1) collected data during the financial distress period of Vietnam; (2) small sample size; and (3) data for calculating yearly dividend payout is not always provided so that the independent variable is not truly reliable.

From the above, we can see that the impact of dividend policy on a firm's performance has been investigated by many scholars from both developed countries and developing countries. And the results are not consistent. Some authors believe that dividend affects a firm's performance significantly and positively, while others show the reversed results, and some scholars indicate that there is no relationship between these two factors.

From theories and literature review there are two research hypotheses proposed as following:

**Hypotheses 1 (H1).** *Dividend rate has negative impact on firms' financial performance.*

**Hypotheses 2 (H2).** *Decision of dividend payment has positive impact on firms' financial performance.*

## 3. Research Methodology

### 3.1. Research Model

From the literature review of previous research models, we found that most of the studies have measured the impact of dividend policy on the firm's financial performance. The following research model (Figure 1) is built on Tobin's Q, ROA, and ROE as a measurement of the firms' financial performance and five variables as independent variables. This measure is consistent with the studies of previous scholars such as Amidu (2007), Murekefu and Ouma (2012), and Velnampy et al. (2014), Dogan and Topal (2014), Tran et al. (2015), and Khan et al. (2016).

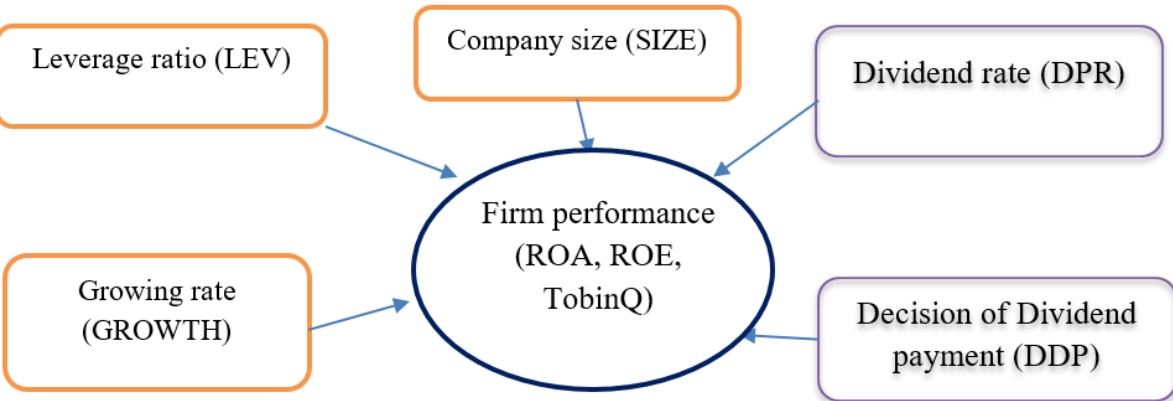

**Figure 1.** The proposed research model.

From the general model, we decomposed it into three separate models for each dependent variable, as follow:

(1) $\text{ROA}_{it} = \beta_0 + \beta_1 \text{DPR}_{it} + \beta_2 \text{DDP}_{it} + \beta_3 \text{SIZE}_{it} + \beta_4 \text{LEV}_{it} + \beta_5 \text{GROWTH}_{it} + \varepsilon_{it}$

(2) $\text{ROE}_{it} = \beta_0 + \beta_1 \text{DPR}_{it} + \beta_2 \text{DDP}_{it} + \beta_3 \text{SIZE}_{it} + \beta_4 \text{LEV}_{it} + \beta_5 \text{GROWTH}_{it} + \varepsilon_{it}$

(3) $\text{TOBIN'SQ}_{it} = \beta_0 + \beta_1 \text{DPR}_{it} + \beta_2 \text{DDP}_{it} + \beta_3 \text{SIZE}_{it} + \beta_4 \text{LEV}_{it} + \beta_5 \text{GROWTH}_{it} + \varepsilon_{it}$

where:

$\text{ROA}_{it}$: Return on average total assets of the company i period t

$\text{ROE}_{it}$: Return on average equity of company i period t.

$\text{TOBIN'S Q}_{it}$: is a measure of firm assets in relation to a firm's market value (the company i, period t). The formula for Tobin's Q is: Tobin's Q = Total Market Value of Firm/Total Book Value of Firm

$\text{DPR}_{it}$: dividend rate (percent) of the company i period t. DPR measured by the amount of dividend on par value of share.

$\text{DDP}_{it}$: Decision of dividend payment of the company i period t. It takes value of 1 if firm pays dividend, otherwise it gets zero.

$\text{SIZE}_{it}$: Logarithm of total assets of the company i period t.

$\text{LEV}_{it}$: Financial leverage of company i period t.

$\text{GROWTH}_{it}$: Growth of revenue of company i period t.

### 3.2. Data Source and Data Collection

As mentioned above, the objective of the research is to investigate the effect of dividend policy (measured by dividend rate, decision of dividend payment) on the firm's performance (ROA, ROE, and Tobin's Q are representative). Thus, we collect financial statements of firms listed on the stock market of Vietnam. We believe that this data source is highly credible because, under Vietnamese law, all listing firms are required to submit audited financial statements. And to overcome the drawback of the previous study, we choose a longer time frame—from 2008 to 2019.

Up to the end of 2019, there are 745 firms listed on Vietnam's official stock exchange. All firms' financial statements from 2008 to 2019 are downloaded. After that, the data is arranged into a multi-column excel file and then ratios for variables for each company in research sample are calculated.

Having estimated the ratios, we check the data and found out that several companies did not disclose the dividend rate and/or decision of dividend payment. Besides, some of the listed companies have a data span of less than 12 years. These companies are removed from the research sample. At the end, only 450 eligible firms to be selected for the research study.

According to Tabachnick and Fidell (1996), the sample size should be n = 50 + 8*m (where *m* is the number of variables). Based on this, it is believed that the research sample is satisfied for running a statistical test or regression and research results may be applied to the whole population.

### 3.3. Data Processing Method

Due to the use of pooled data, we have to apply the appropriate method of processing the data. Various statistical and econometric methods and techniques are applied step-by-step, as follows:

First, data descriptions by min, max, median, mode, and standard deviation. This will provide some general features of the firms such as size, growth, leverage, cash dividend per share, the proportion of earnings for the dividend. The statistical description also reports the variance between firms relating to each variable in the research model.

Second, a correlation test is conducted to check the relationship between independent variables and dependent variables. If two independent variables are strongly correlated, reflecting a reasonably perfect value of correlation coefficient (around 1.0), the research model may have a multicollinearity problem, in which case one independent variable must be removed. Similarly, if the correlation coefficient between independent variables and the dependent variable is zero, this means that there is no correlation between them. As a result, that independent variable is not suitable for the research model.

Third, multicollinearity problem is tested. Multicollinearity exists whenever an independent variable is highly correlated with one or more of the other independent variables in a multiple regression equation. Multicollinearity is a problem because it undermines the statistical significance of an independent variable. It is advised that the VIF predictor should be calculated. If VIF is greater than 5, it means multicollinearity problem exists (Hoang and Chu 2013).

Fourth, the F-test is used to choose the best fit model among OLS and FEM. If FEM is chosen, the Hausman test is conducted to choose between FEM or REM.

Finally, beta for independent variables in the model is tested. By doing this, the variables which have the greatest effect on firms' performance could be found and whether the influence is positive or negative at a specific statistical significance level. The results of the tests are presented in the following section.

### 4. Empirical Findings

The statistical description in Table 1 indicates that the sample firms have an average ROE of 14 per cent, average ROA of around 7 per cent, and average Tobin's Q is only 0.61—the market value of listing firms is lower than to their book value. The low Tobin's Q implies the pessimistic about the firms' future development.

For the dividend variables, the results indicate that there are more than 60 per cent of sample firms paid dividend to shareholders (measured by DDP). However, the average dividend rate for the sample firms is quite low, around 10.5 per cent.

The second test conducted is the correlation between variables and the result is reported in Table 2, as follows.

**Table 1.** Statistical description of the research model.

| Variable | Obs | Mean | Std. Dev. | Min | Max |
|----------|-----|------|-----------|-----|-----|
| ROE | 5400 | 0.140 | 0.1624 | −1.750 | 2.930 |
| ROA | 5400 | 0.069 | 0.087 | −0.900 | 0.810 |
| TobinQ | 5400 | 0.612 | 0.722 | 0.010 | 7.260 |
| Size | 5400 | 27.019 | 1.495 | 21.150 | 33.630 |
| Growth | 5400 | 14.890 | 64.606 | −100.000 | 967.15 |
| Lev | 5400 | 50.590 | 22.030 | 0.350 | 203.060 |
| DPR | 5400 | 10.459 | 13.296 | 0.000 | 214.000 |
| DDP | 5400 | 0.614 | 0.487 | 0.000 | 1.000 |

**Table 2.** Correlation matrix between independent variables.

| | Size | Growth | Lev | DPR | DDP |
|---|------|--------|-----|-----|-----|
| Size | 1.000 | | | | |
| Growth | 0.065 | 1.000 | | | |
| Lev | 0.299 | 0.026 | 1.000 | | |
| DPR | 0.034 | −0.039 | −0.161 | 1.000 | |
| DDP | 0.028 | −0.053 | −0.070 | 0.624 | 1.000 |

The results from Table 2 show that the dividend policy variables (measured by DPR and DDP), have a positive relationship to the firms' performance measured by ROA, ROE, and Tobin Q. Conversely, leverage and firm size (measured by logarithms of assets) have a negative correlation to firms' performance. The study results also show that the independent and dependent variables are correlated with each other, satisfying correlation conditions, and no variable is removed from the research model.

We also check for multicollinearity problems using the VIF indicator. The results reporting in Table 3 show that the VIF is smaller than 2, meaning there no multicollinearity issues in our research model (Hoang and Chu 2013).

**Table 3.** Result for multicollinearity phenomenon.

| Variable | VIF | 1/VIF |
|----------|-----|-------|
| DPR | 1.68 | 0.593976 |
| DDP | 1.64 | 0.609223 |
| Lev | 1.14 | 0.879604 |
| Size | 1.11 | 0.899836 |
| Growth | 1.01 | 0.992749 |
| Mean VIF | 1.32 | |

Autocorrelation is also tested by using Wooldridge test in panel data. The test result is

$$F(1, 449) = 2.149; \text{Prob} > F = 0.1434$$

This test result implies that there is no serial autocorrelation in the model.

After checking for correlation, autocorrelation, and multicollinearity problems, we continue evaluating the appropriateness of the regression model for panel data. To do this, we first conduct the Pooled OLS and Fixed effect model (FEM). The results from F-test show that FEM is more appropriate than OLS. This leads to the test being performed to compare between FEM and Random effect model (REM) by using the Hausman test, and we found

that the FEM is more appropriate and chosen. The regression results are presented in the following Table 4.

**Table 4.** Regression results for three dependent variables.

| | Indicators | ROA | ROE | TobinQ |
|---|---|---|---|---|
| | Coefficient | 0.0008 | 0.0015 | −0.0021 |
| DPR | *t*-value | −4.84 | 6.77 | −5.86 |
| | Sig. level | 0.000 | 0.000 | 0.000 |
| | Coefficient | −0.0085 | −0.0183 | 0.0189 |
| DDP | *t*-value | −3.48 | −3.16 | 1.93 |
| | Sig. level | 0.001 | 0.002 | 0.053 |
| | Coefficient | −0.0080 | −0.0323 | −0.0985 |
| Size | *t*-value | −4.84 | −8.29 | −15.01 |
| | Sig. level | 0.000 | 0.000 | 0.000 |
| | Coefficient | 0.0002 | 0.0004 | −0.0001 |
| Growth | *t*-value | 15.15 | 13.96 | −2.23 |
| | Sig. level | 0.000 | 0.000 | 0.026 |
| | Coefficient | −0.0014 | 0.0001 | −0.0082 |
| Leverage | *t*-value | −17.28 | 0.080 | −26.44 |
| | Sig. level | 0.000 | 0.938 | 0.000 |
| | Coefficient | 0.3460 | 1.001 | 3.704 |
| Con. | *t*-value | 7.97 | 9.71 | 21.33 |
| | Sig. level | 0.000 | 0.000 | 0.000 |
| | F-statistic | 133.12 | 60.21 | 250.37 |
| Model | *p*-value (F-statistic) | 0.000 | 0.000 | 0.000 |
| | R-squared | 0.2217 | 0.0574 | 0.1364 |

The research model for ROA could be presented as the following equation:

$$ROA_{i,t} = 0.346 + 0.0008DPR - 0.0085DDP - 0.0080SIZE - 0.0014LEV + 0.0002GROWTH + \varepsilon_{it}$$

First, the dividend rate (DPR) had a positive effect on ROA, statistically significant at 1% level. This result indicates that a higher rate of dividend leads to a higher return on total assets, but the influence level is very small. When the dividend rate increases by 1 percent, the ROA increases only 0.0008 percent. This result is similar to the study by Amidu (2007) but in contrast to the study by Khan et al. (2016), who evidenced that dividend rate is negatively correlated to ROA. This phenomenon may be explained by the fact that when companies increase dividend, they can mobilize more capital for further development, and, as a result, the profitability decreases.

Second, decision of dividend payment (DDP) has negative effect on ROA, at statistical level of 1%. It implies that the declaration of dividend affects to firm's performance in negative manner.

Third, with regard to the controlling variables, both firms' asset size (SIZE) and leverage (LEV) have a statistically negative effect on ROA. However, the firms' sale growth (GROWTH) had a significantly positive impact on ROA.

Fourth, the regression results also show that the value of R-square = 0.2217, which means that the independent variables of the model explained 22.17% of the change of the dependent variable.

The research model for ROE could be presented as the following equation:

$$ROE_{i,t} = 1.001 + 0.0015 * DPR - 0.0183 * DDP + 0.0004 * GROWTH - 0.0323 * SIZE + 0.0001 * LEV + \varepsilon$$

First, similarly as ROA, the DPR variable has a positive impact on ROE at a significance level of 1% and the influence level is also small. Specifically, the increase of DPR by 1% leads to an increase of ROE by 0.010%.

Second, DDP has a negative impact on ROE at statistically significant level of 1%. As same as ROA, firms' declaration of dividend impact reversely to firms' financial performance.

Third, for controlling variables, the result shows that only sale growth and firm size have statistically significant impact on ROE in different ways, sale growth leads to higher performance but firm size results in lower performance. Surprisingly, the firms' leverage does not have significant affect to ROE.

Fourth, the regression results also show the value of R-square = 0.0574, which indicates that the independent variables of the model explain only 5.74% of the change of the dependent variable. There are many other factors affecting to return on equity.

For the Tobin's Q, the research model could be presented as the following equation:

$$TOBIN'SQ_{i,t} = 3.704 - 0.0021 * DPR + 0.0189 * DDP - 0.0001 * GROWTH - 0.0985 * SIZE - 0.0082 * LEV + \varepsilon$$

The research results indicate that:

First, DPR negatively impacts Tobin's Q a statistically significance level of 1% with small magnitude. It means that if the dividend rate increases by 1 percent, Tobin's Q will decrease by 0.0021 percentage point. The decision of dividend payment (DDP) also contributes to the increase of Tobin's Q, at a significant level of 10%.

For controlling variables, financial leverage, sales growth and size have negative impact on Tobin's Q and they are all statistically significant.

For the appropriation of the model, the regression results also show that the value of R-square = 0.1364 which shows that the independent variables of the model could explain 13.52% of the change in Tobin's Q.

From the results of the above three models, we could realize that both in book value and market value, the dividend rate (DPR) has an inverse impact on firms' financial performance, meanwhile the decision of dividend payment (DDP) has diverse effects on firm's performance.

## 5. Conclusions and Recommendations

Based on the results of the research above, it can be seen that the Vietnamese firms offer quite low dividend rate, an average amount of 10 per cent. This low dividend rate, in one hand, evidence that firms are retaining profit for operation and future investment. As a result, some shareholders believe on future prospect so that they continue investing their money into the firms. However, the low dividend, on the other hand leads to the decrease of market expectation to the firms' future development.

The high deviation value (16.3%), the wide variance range of dividend rate (0 to 214 per cent) may reflect the lack of systematic and reasonable dividend policy in Vietnamese listed firms. Firms may subjectively offer a high dividend rate or high cash dividend for the year in which they want to mobilize capital without taking into consideration the effect of such policies on the firms' financial performance, either the accounting-based value or market-based value.

Based on the research results, the paper proposes the following instructive suggestions:

First, firms should choose and apply the economic model for dividend policies, including dividend rate, decision of dividend payment. Model of dividend policy should be stable, long-term, strategic, and not be affected by immediate influence of firms' managers. By doing that, the firms may control the cashflow and the required capital structure to achieve the best financial performance.

Second, firms' dividend policies should embed investment policy and financing policy for each stage in firm life cycle. For instance, in introduction and growth stages, firms should have low dividend rate but in the maturity period, where high profit and cash available, the dividend should be high.

Third, it is evidenced that dividend policy also depend on diverse factors. Therefore, firms should take into account the factors like country characteristic, development period, and on agency cost of debt. These suggestions have been evidenced by several scholars (for instance, Brockman and Unlu 2009) or country culture (Zheng and Ashraf 2014).

Fouth, firms need to realize that the decision of dividend payment (DDP) has a negative impact on firms' financial performance, and dividend rate impact positively to firms' performance. Thus, firms should construct progressive strategy of dividend payout rate, with one steady rate and some additional payments for special circumstances.

And last but not least, firms should clearly communicate to shareholders about the tradeoff between high dividend payout rate and performance, so that shareholders will willingly accept the low dividend rate, retaining profit as capital for future development.

Research on the impact of dividend policies on the financial performance of Vietnamese listed enterprises evidence that the dividend rate and decision of dividend payment have an effect on firms' financial performance, measured by ROA, ROE, and Tobin's Q in different ways. Based on the findings, we also suggest some instructive recommendations for firms, including a more appropriate model of dividend policies, keeping low dividend rate, and clear declaration of dividend payment. These might be useful for listed firms, regulators, investors, and others in making investment decisions for firms.

However, the research model may still contain some limitations. One of this is related to low R-square in the model. It means that the paper may have not uncovered many other factors related to dividend policies that directly or indirectly affect the firm's performance. In addition, the paper has not taken into account the effect of time, industry, firm's age, etc. We believe that those potential drawbacks may be the gap for further research in the future.

**Author Contributions:** Conceptualization: C.D.P., A.H.N. Methodology: C.D.P. Software: T.V.T. Validation: H.T.N. Formal Analysis: T.T.T. Investigation: T.V.T. Data curation: N.T.D., T.T.T. Writing original draft preparation: T.T.T. Writing, Review and Editing: C.D.P. Supervision: C.D.P. Final Acquisition: A.H.N., H.T.N. All authors have read and agreed to the published version of the manuscript.

**Funding:** This research received no external funding.

**Institutional Review Board Statement:** The study was conducted according to the guidelines of the Declaration of Helsinki, and approved by the Institutional Review Board.

**Informed Consent Statement:** Informed consent was obtained from all subjects involved in the study.

**Data Availability Statement:** Data used for this study is collected from audited financial statements of Vietnamese firms which are listing in the stock market of Vietnam.

**Acknowledgments:** This research is funded by the National Economics University (NEU), Vietnam. The authors thank anonymous reviewers for their contributions and the NEU for supporting this research.

**Conflicts of Interest:** The authors declare no conflict of interest.

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
