# Peer review of "The Effect of Dividend Payment on Firm’s Financial Performance: An Empirical Study of Vietnam"

_jrfm, doi:10.3390/jrfm14080353_

Round 1
Reviewer 1 Report
The strengths of the article is original and interesting considerations with is consistent with the pattern of research. Solid methodology of the research with statistical analysis.
Therefore contribution to existing knowledge is considerable. Also advantage of the research is perfect organization & readability.
I cannot find the weaknesses of the assessed article.
Model article worthy of imitation.
In generally it is excellent article and very interesting considerations, which is consistent with the pattern of research. A very good review article with the analysis of statistics on the topic under study.
Author Response
We thank you for your comment and advice
Reviewer 2 Report
The paper aims to investigate the effects of dividend policies on a firms’ financial performance. The paper explores the research gap and then builds a research model using ROA, ROE, and Tobin’s Q as dependent variables, dividend rate and decision of dividend payment as independent variables. The topic is interesting and within the scope of the journal.
With a few exceptions, the paper is well structured and written.
The abstract is clear, it presents the object of research and the results.
The results and interpretations are correct but they should refer to the results of previous studies.
Major concerns
In the introduction, the authors should explain why Vietnamese SMEs are worth studying. Also, the main results have to be presented.
The literature review section should be improved with papers focused on emerging economies.
You should clarify the contributions of the paper which are not elaborated well in the current paper. You can talk about the following contributions: What insights can you provide based on your finding? Do they push forward our understanding? What should we do with your research? Do you have any suggestions to improve the current regulation or practice? Adding the above discussion and extend your literature review may help you make more contributions and position your contributions better.
The practical and policy implications of your findings have to be presented. Also, the limits of your study in the conclusions section.
How do you treat endogeneity issues in your empirical model?
Suggested readings:
Anton, S.G. (2016). The impact of dividend policy on firm value. A panel data analysis of Romanian listed firms, Journal of Public Administration, Finance and Law, https://www.ceeol.com/search/article-detail?id=743810
In conclusion, I would like to thank the authors for a very interesting, unique, and potentially important paper. It was a pleasure to read this manuscript. Hope these comments and suggestions can help further their study. I wish the authors the best.
Reviewer 3 Report
The paper is interesting and has a publication potential, but several issues need to be addressed and fixed:
"A dividend is a distribution of profits by a corporation to its shareholders. When a corporation earns a profit" this looks like a definition. It should be properly referenced.
"instance, Gill at al., 2010" it is "et al."
The introduction should be enhanced: the research gap is not that clear highlighted. The way how the paper adds value to the literature and the connection research scope/question - novelty - enhanced theory should be better emphasized.
In the literature review section you should have at least one reference in each paragraph. Furthermore you should also try to have newer references. It is nice to start with a reference from 1961, but I would expect that you say something like ... although these aspects have been investigated starting with 1961, the newer developments show this and this ...
75% of the references should be from the last 2-3 years. Your references are quite old...
The proposed research model has 5 arrows, so for each arrow you should have a hypothesis in the lit review. You have only 2. Why? There should be 5. You should also explain in the lit review the 6 different constructs of your research model
"on Vietnam’s official stock exchange" is there also a unofficial stock exchange?
for section 3.3. please also cite some references that state that this way of conducting the research is the proper one.
The paper is missing a discussions section where own results are compared to previous findings of the international literature, so that the originality of the paper is shown.
The conclusions should contain 4 paragraphs:
- theoretical implications/ how does the paper extend the literature; 2. managerial implications; 3 limitations; 4. Future research perspectives
Please do not have paragraphs consisting of only one phrase.
The conclusions should be stronger. They are quite weak.
Reference list should be extended. More references should be cited. New references must be included in the paper.
Round 2
Reviewer 2 Report
Dear Authors,
I have carefully read the revised version of the paper The Effect of Dividend Payment on Firm’s Financial Performance: An Empirical Study of Vietnam and I checked your responses. I consider that you have improved significantly the paper according to the reviewers’ recommendations. Thus, I consider that the paper can be published in this form. Congrats!
Best regards
Reviewer 3 Report
As most of my suggestions were implemented the paper can be accepted. Best luck with attracting citations.
This manuscript is a resubmission of an earlier submission. The following is a list of the peer review reports and author responses from that submission.
Round 1
Reviewer 1 Report
This paper examines the relationship between dividend policy and firm performance in the context of Vietnam. I have key concerns about the suitability of the paper for JRFM. These are as follows:
1. Motivation: It is not clear what is the motivation of the paper. On p. 6, the authors mention that a similar study has been conducted by Tran et al. (2015). While the current study sample is higher than the previous study, is the result really different? Is the result different if 2009-2013 period is excluded from the sample?
2. Variable definitions: It is not clear whether stock dividend is included in the calculation of DPR. From Section 5, it seems like yes. If so, stock divided should not be included in the calculation as it does not represent distribution of profits.
3. Models: In their IJFR paper, Tran and Nguyen (2014) show that partial adjustment model is consistent with Vietnamese stock market. for that, the models should control for 1-year lagged dividend in the model.
Since the study uses multiple years of data and as the results could change due to industry characteristics, there should be controls for year and industries.
4. Results: DPR has a maximum value of 1.040, meaning that the firm has paid dividend in excess of profits earned. Is it allowed in Vietnam to pay dividend from retained earnings?
The minimum values of ROE and ROA is 0, meaning that none of the firms in the sample incurs any loss during the sample period. It also indicates that the sample selection process is biased as it does not include loss making firms. So the results can't be generalized across all listed firms.
It is not clear why DPR and DPS should take opposite signs in the regression model.
The R-sq value is quite low compared to other studies cited. it means the models suffer from the omitted variable problems.
Endogeneity is a common concern is this type of research where the performance of the firm is very likely to influence the dividend payout ratios. There is no discussion on this issue in the paper.
Reviewer 2 Report
The strengths of the article is original and interesting considerations with is consistent with the pattern of research. Solid methodology of the research with statistical analysis.
Therefore contribution to existing knowledge is considerable. Also advantage of the research is perfect organization & readability.
I cannot find the weaknesses of the assessed article. Model article worthy of imitation.
In generally it is excellent article and very interesting considerations, which is consistent with the pattern of research. A very good review article with the analysis of statistics on the topic under study.
Overall evaluation: article it is suitable for publication in current version.
Reviewer 3 Report
The paper aims to investigate the effect of dividend policy (measured by dividend per share, dividend payout) on the firm’s performance (with ROA, ROE, ROIC, and Tobin’s Q are representative). The topic of the manuscript is relevant to the scope of JRFM.
With a few exceptions, the paper is poorly structured and written.
The methodology is poor. The FE doesn’t take into account the endogeneity which is a serious concern in corporate finance literature.
The results and interpretations are correct, but they should refer to the previous results.
The aim of the paper is not well defined in the Introduction.
The authors must stress why is it worth studying Vietnam and not India, for example? Here, you should relate the discussion with the previous literature on emerging markets.
A literature review section should be included. You can take some paragraphs from the Introduction.
There are numerous typos in the paper need to be addressed.
The authors should highlight the limits of their research. The last section of conclusion should summarize all your findings, their implications to researchers and practitioners, the future direction for research, limitation of the current study, etc.
You should clarify the contributions of the paper which are not elaborated well in the current paper. You can talk about the following contributions: What insights can you provide based on your finding? Do they push forward our understanding? What should we do with your research? Do you have any suggestions to improve the current regulation or practice? Adding the above discussion and extend your literature review may help you make more contributions and position your contributions better.
Best Regards,